# Ultrasonographic Assessment of Caudal Vena Cava Collapsibility Index, Caudal Vena Cava-to-Aorta, and Femoral Vein-to-Artery Ratios in Healthy Sedated Adult Horses

**DOI:** 10.3390/ani15192837

**Published:** 2025-09-28

**Authors:** Elisa Scala, Inge Durie, Kris Gommeren, Claude Saegerman, Gaby van Galen

**Affiliations:** 1Evidensia Strömsholm Referral Equine Hospital, 734 94 Strömsholm, Sweden; 2Department of Clinical Sciences, Faculty of Veterinary Medicine, University of Liege, 4000 Liege, Belgium; 3Department of Infections and Parasitic Diseases, Faculty of Veterinary Medicine, University of Liege, 4000 Liege, Belgium; 4Sydney School of Veterinary Science, University of Sydney, Sydney, NSW 2006, Australia; 5Goulburn Valley Equine Hospital, Congupna, VIC 3633, Australia

**Keywords:** equine, ultrasound, blood vessels, transrectal, inguinal

## Abstract

This is a prospective, observational pilot study conducted at a private equine referral hospital that aims to establish baseline data in adult horses for ultrasonographic evaluation of the caudal vena cava respiratory collapsibility index and large vessel vein-to-artery ratios, which are currently used in human and small animal emergency medicine to guide fluid therapy. This study showed that both transrectal and inguinal ultrasonographic approaches are well-tolerated and feasible and offers valuable insights for the future development of non-invasive tools to assess intravascular volume status and fluid responsiveness in horses.

## 1. Introduction

Cardiovascular function and tissue perfusion are often impaired in critically ill equine patients, especially in cases of gastrointestinal disease, sepsis, or hemorrhage [1,2]. Accurate assessment of volume status and fluid responsiveness is essential to guide therapy. In other species, traditional methods—clinical signs, laboratory tests (hematocrit, total protein, venous L-lactate), and blood pressure measurements—are considered indirect and inaccurate in assessing volume status [3,4] and do not predict fluid responsiveness [5], which is influenced by factors such as cardiac function, vascular permeability, and the glycocalyx [5]. Poor responders to fluid therapy show a limited increase in cardiac output [5,6] and are more prone to fluid overload, which may reduce oxygen delivery and worsen tissue hypoxia [5,6]; these patients may rather benefit from inotropic or vasopressor support [5,6].

In human medicine, ultrasonographic assessment of the inferior vena cava (IVC) provides a dynamic evaluation of volume status [7,8]. In adults, IVC diameter and respiratory-related variation are routinely measured [7,8]. At end-expiration, the IVC reaches its maximum (max) diameter because of reduced venous return due to increased intrathoracic pressure; at end-inspiration, it reaches its minimum (min) in spontaneously breathing patients [7,8,9]. This allows calculation of the respiratory collapsibility index (CI) [7,8,9]. The IVC CI exceeds 50% in hypovolemia, while in hypervolemic patients a dilated IVC shows little to no collapse [7,8,9]. The IVC-to-Ao ratio (IVC/Ao) can also be used, with values under 1.14 indicating hypovolemia [10]. Femoral artery (FA) and vein (FV) ultrasonography and the FV/FA ratio have also been proposed to have potential in this matter [11,12]. Reduced diameter of these vessels indicates low venous pressure (FV) or systemic vasoconstriction (FA). Besides assessing volume status, IVC CI and IVC/Ao have been used to assess fluid responsiveness [7,8].

In recent years, multiple studies on this topic have been published in veterinary medicine. In dogs, respiratory-related variation in the caudal vena cava (CVC) has been observed at or near the diaphragm [6,13,14,15,16], while no variation has been identified medial to the kidneys [14,17]. Similar findings have been reported in healthy foals and calves [18,19,20]. The CVC collapsibility index (CI CVC) shows potential in clinical settings in dogs and foals [6,13,15,18,21], as does the CVC-to-aorta ratio (CVC/Ao) in dogs, foals, and calves [15,17,18,19,20,22,23,24,25,26].

To the best of the author’s knowledge, there are currently no studies performed in adult horses on ultrasonographic evaluation of large blood vessels for the purpose of volume status assessment or fluid responsiveness. This study aimed to determine whether ultrasound can be used to examine the CVC, aorta (Ao), FV, and FA in healthy, sedated horses using transrectal and transcutaneous inguinal techniques. More specifically, this study involved the following: (1) describing the ultrasonographic procedure, encountered complications, vessel anatomy, and measurements; (2) calculating the respiratory CI and vein-to-artery ratios; (3) comparing measurements taken in B- and M-mode; and (4) evaluating repeatability and reproducibility. We hypothesized that ultrasonographic evaluation of these vessels and related indices would be feasible, repeatable, reproducible, and consistent across imaging modes.

## 2. Materials and Methods

This was a prospective, observational, single-center proof-of-concept study. Data was collected between 1 February and 23 March 2023. Ethical approval was obtained from the Swedish Board of Agriculture (Jordbruksverket; Dnr 5.2.18-01302/2021, dated 8 June 2021, and Dnr 5.8.18-22604/2021, dated 28 January 2022). Owner informed consent was provided.

### 2.1. Animals

Seventeen horses were recruited from the hospital staff and included privately owned adult animals. No formal sample size calculation was performed due to the exploratory nature of this pilot study and taking into account practical considerations such as time and resource availability. Examinations were performed at the Evidensia equine hospital in Strömsholm (Sweden) or at the farm. Information on signalment, body weight (obtained using an electronic horse scale at the clinic or a weight tape on the farm), and withers height (measured with a horse measuring stick) was collected. The study included 9 mares and 8 geldings: 10 Warmbloods, 6 Standardbreds, and 1 Anglo-Arab mixed breed. Mean age was 9 ± 5 years, mean height was 162 ± 7 cm, and mean weight was 551 ± 50 kg with a body condition score [27] ranging from 4 to 7/9.

The inclusion criteria were an age over three years,, a withers height exceeding 140 cm, and judged healthy based on full clinical examination and blood analysis (packed cell volume (PCV) and blood L-lactate), performed prior to sedation and ultrasonographic examinations. Blood L-lactate was measured using the StatStrip Xpress Lactate System with StatStrip Lactate Test Strips (Nova Biomedical, Waltham, MA, USA), while PCV was determined by centrifugation or by the automated hematology analyzer Sysmex XN-1000 (Sysmex Corporation, model XN-1000, Kobe, Japan). Exclusion criteria encompassed any clinical signs of hypovolemia or systemic disease, including heart rate > 80/min, blood lactate > 2 mmol/L, PCV < 0.25 or > 0.5 L/L, or rectal temperature > 38.3 °C. Two horses were not included in the study prior to ultrasonographic examinations: one horse presented with an elevated rectal temperature, while the other one collapsed after inadvertent arterial injection of xylazine. Both horses recovered uneventfully.

To allow for a safe ultrasonographic examination, all horses were sedated with xylazine (0.6 mg/kg IV; Xysol vet. 20 mg/mL; VM Pharma AB, Stockholm, Sweden) prior. The level of sedation was estimated by the operators as mild when the horse exhibited slight ptosis and a reduced, delayed response to environmental stimuli and as moderate when characterized by marked ptosis, absence of response to environmental stimuli, and ataxia. The ultrasonographic examinations were commenced when the horse showed signs of sedation, typically around 5 min after xylazine administration. If the examinations were performed at the clinic, the horses were also restrained in a stock; otherwise, examinations were performed with no additional restraint.

### 2.2. Ultrasonographic Technique and Image Acquisition

The abdominal Ao and CVC were examined via transrectal ultrasonography, while the FA and FV were examined transcutaneously in the inguinal area (further referred to as inguinal).

Two operators (OP1 and OP2) performed the ultrasound procedure twice to evaluate intra-rater variability (determined by calculations of reproducibility) and inter-rater variability (determined by calculations of repeatability) (2× OP1; 2× OP2). Each ultrasonographic exam was timed from first probe contact on the animal until contact was lost. Operator 1 (OP1: ES) was, at the time of the experiment, a final-year equine internal medicine resident with moderate experience in ultrasonography, while operator 2 (OP2: ID) was an established equine internal medicine specialist (>10 years) with extensive ultrasonographic experience.

Ultrasonographic loops of 30 s duration were recorded of the transverse view of all vessels in B- and in M-mode. Minimal pressure was applied to avoid iatrogenic compression of the vessels during ultrasonography. The operator who was not examining the horse timed the beginning of inspiration to allow the other operator to start the recording at the appropriate time. Both operators were therefore present throughout the entire study (one timing respiration, one scanning). Operators were not allowed to comment on or view each other’s ultrasonographic images during examinations. Additionally, each operator’s exam was saved as a separate ultrasonographic study with a blinded and randomly assigned file name.

Degree of sedation and complications or difficulties encountered during the examinations were recorded.

To assess the feasibility of the technique, one horse was scanned twice by OP1 prior to the study.

#### 2.2.1. Transrectal Ultrasonographic Technique

Rectal ultrasonography (MyLab™ DeltaVET, Esaote S.p.A., Genoa, Italy) was performed using a 5 MHz rectal probe set at a maximum depth of 6.5 cm and focus at 3.5 cm. To allow for a safer rectal examination, 60 mL of local anesthetic was instilled intrarectally in all horses right after sedation. Lidocaine (20 mg/mL; Lidokain Mylan, injection solution, 20 mg/mL; Mylan Ireland Limited/SALF SpA Laboratorio Farmacologico, Bergamo, Italy) was administered to the first two horses, but due to inadequate relaxation, mepivacaine (10 mg/mL; diluted Carbocain^®^ injection solution, 20 mg/mL; Aspen Nordic, Ballerup, Denmark) was used for the remaining cases. Around 5 min after sedation, feces were evacuated from the rectum, and right-handedly, the hand was positioned to scan dorsally on a sagittal or rightward plane, approximately 50–60 cm cranial to the anus, and the ultrasound transducer was angled at 45° towards the right cranial abdomen (Figure 1) to visualize the Ao and CVC. The bifurcation of the Ao was identified, after which a window of the Ao and CVC approximately 2 cm cranial to this bifurcation (Figure 2) was selected. Care was taken to optimize a transverse view of the CVC and the Ao at this location, which was then assessed in B- and M-mode for each view. In between each operator’s transrectal ultrasonographic examination, the hand was retracted to the caudal rectum, but not out of the rectum, to reposition the probe while avoiding rectal irritation and air buildup.

#### 2.2.2. Transcutaneous Inguinal Ultrasonographic Technique

Inguinal ultrasonography was performed with the same device (MyLab™ DeltaVET, Esaote S.p.A., Genoa, Italy), using an 8–10 MHz linear probe set at a maximum depth of 2–3.5 cm and a focus of 1.5 cm. The left femoral triangle was clipped and cleaned, and alcohol and/or gel was applied on the skin to enhance contact. For safe examination, the operators positioned themselves on the left side of the horse, and the ultrasound transducer was positioned left-handedly on the inner left hind limb to visualize a transverse view of the FA and FV in subcutaneous location, as proximally as possible (Figure 3).

### 2.3. Ultrasonographic Measurements and Indexes Performed After Image Acquisition

All ultrasonographic recordings were reviewed blindly by OP1 in random order, 5 to 6 months after the last ultrasonographic examination took place, with no access to the horse, operator, or examination details of the relative ultrasound images. The program RadiAnt DICOM Viewer (version 2023.1, 64-bit; Medixant, Poznań, Poland) was used to analyze the recordings.

#### 2.3.1. General Variables

The ultrasonographic image quality was graded on a scale from 1 to 3: (1) severe artifacts preventing full visualization or precise delineation of one or more vessels throughout the entire loop; (2) moderate artifacts partially affecting vessel delineation, such as blurring of one edge or occurring during a part of the recording; (3) high-quality image with no artifacts.

All diameters were measured from the distal far-field to the proximal border in the near field (Figure 4). Where the vessel had an irregular shape (CVC), the diameter was measured where the vessel’s walls were as close as possible to being perpendicular to the ultrasound beam. All areas were measured following the internal lining of the vessels’ wall. During measurement, the ultrasonographic loops were subjectively categorized as either easy or difficult to measure, based on how easy it was to identify the optimal moment for measuring in the ultrasonographic recording—especially considering the effect on caudal vena cava size during expiration and inspiration (respectively, maximum and minimum diameter and area).

#### 2.3.2. Transrectal Variables

Diameter (cm) and area (cm^2^) of the caudal vena cava (CVC, min and max) and aorta (Ao, systole) were measured in B- and/or M-mode from transrectal loops (Figure 4; Table 1).

The Ao was measured only in systole, when the vessel presented a clear pulsation and the diameter was at its largest; the Ao area was calculated from the diameter. CVC max and min values were selected visually, considering that recordings began at inspiration onset—thus, min values correspond to end-inspiration and max values to end-expiration.

CI CVC (Table 1) was calculated for diameter or area as follows:$$CI\;CVC=\frac{CVC\;max-CVC\;min}{CVC\;max}$$

CVC/Ao (Table 1) was instead calculated as follows:$$CVC/Ao=\frac{CVC\;max}{Ao}$$

#### 2.3.3. Inguinal Variables

The B-mode diameter (cm) and area (cm^2^) of FA and FV were measured at their maximum value to prevent underestimating vessel size due to external pressure from the operator or the ultrasound transducer (Figure 4; Table 1).

The FV/FA (Table 1) was calculated for diameter or area as follows:$$FV/FA=\frac{FV}{FA}$$

### 2.4. Statistical Analysis

For data normality evaluation and statistical comparison, all measurements were pooled without accounting for repeated measures on the same individual (i.e., the data were treated as independent observations).

Data normality was evaluated with the Shapiro–Wilk test. Mean and standard deviation (SD) or median and interquartile range (IQR) were calculated for each ultrasonographic measurement for normally and non-normally distributed data, respectively.

Statistical comparison for CVC max versus CVC min measurements and measurements and calculations in B-mode versus M-mode were performed by a two-sample Wilcoxon rank-sum (Mann–Whitney) test. Statistical significance was set at a *p*-value of less than 0.05.

To assess agreement for derived indices, repeatability (same operator, two measurements) and reproducibility (two operators, same measurement) were evaluated for the following ratios: pooled data of CI CVC, CVC/Ao (based on B-mode and M-mode diameter and area) for transrectal variables, and pooled data of FV/FA (based on B-mode diameter and area) for inguinal variables. The analysis followed the method described by Watson and Petrie [30]. For each pair of observations, the difference was calculated, followed by the mean and standard error (SE) of these differences. Assuming a normal distribution of differences, 95% of all differences in the population are expected to lie within the range of the mean ± 1.96 × SE. The disagreement limit was considered to be values over 5%. A Bland–Altman plot was subsequently generated based on the differences between each pair of observations, using pooled data of CI CVC, CVC/Ao (transrectal variables), and FV/FA values (inguinal variables).

All statistical analysis was performed using commercially available Stata/SE 14.2^®^ (StataCorp LLC, College Station, TX, USA).

## 3. Results

The clinical examination and vital signs were normal in all horses, including pink mucous membranes and mean heart rate, respiratory rate, and rectal temperature of 37 ± 8 bpm, 20 ± 11 bpm, and 37.4 ± 0.4 °C, respectively. The mean packed cell volume was 0.38 ± 0.05 L/L, and the mean blood L-lactate was 0.54 ± 0.28 mmol/L. The level of sedation was estimated by the operators as mild in 8/17 and moderate in 9/17.

### 3.1. Ultrasonography Technique

The median recording time for a single ultrasonographic examination was 4 min (range: 2–12 min) transrectally and 3 min (range: 2–16 min) inguinally. The recording times decreased with experience: for the first two horses, the median recording time was 5 min (range: 2–12 min) for transrectal and 6.5 min (range: 3–16 min) for inguinal. In contrast, for the last five horses, the median time was 3 min for both transrectal (range: 3–7 min) and inguinal ultrasounds (range: 2–4 min).

Ultrasonographic recordings of CVC, Ao, FA, and FV (Figure 4) were successfully obtained in almost all horses (17/17 horses for transrectal examination; 16/17 horses for inguinal). Most horses tolerated the procedures well, but 6 out of 17 showed signs of restlessness before the examination, and 1 during the exam. While sedation was sufficient for the majority, in four cases it wore off too quickly, leading to reactivity and kicking. As a result, one horse was excluded from inguinal ultrasonography by both operators.

Abdominal and inguinal vessels were successfully reached and visualized in transverse view in all horses that could be examined, at least by one operator. For the first two horses in the study, one operator mistakenly captured transrectal images too caudally, at the level of the right internal iliac artery and right common iliac vein. Consequently, these recordings were excluded from the study.

During the transrectal examination, no horse experienced pain or bleeding, but 6/17 were tense, and 8/17 strained. Feces or air accumulation in the rectum was noted in 8/17 horses. All horses were observed by the owner after the examinations and presented no additional discomfort or complications. Complications related to the operators included discomfort and numbness in the operator’s arm, reported in 2 horses during the transrectal examination. Additionally, irregular breathing patterns in some horses made it challenging to synchronize both transrectal and inguinal ultrasonographic recordings with respiration in 3/17. Thirty-second-long ultrasonographic recordings were sufficient to capture a minimum of at least one full respiratory cycle in all horses.

During image processing, 86% (55/64 recordings) of transrectal Ao and 66% (42/64 recordings) of CVC recordings were categorized as easy to measure. Image quality was graded 3 for the majority of the transrectal images (Figure 5). Performing measurements on the M-mode recordings of the FV and FA was considered impossible due to severe artifacts as a result of the vessels’ small size and the horses’ movement. Therefore, these images were not further processed. Inguinal B-mode FA and FV recordings were successfully obtained in 91% of examinations (58/64 recordings). Fewer inguinal images were graded as high quality compared to those obtained transrectally (Figure 5).

### 3.2. Transrectal Variables

In all horses, the Ao displayed clear cardiac pulsations, while the CVC showed only minimal diameter change during the cardiac cycle but larger, rhythmic changes attributed to respiration. The CVC shape was often irregular and fusiform (Figure 2 and Figure 4C–E), especially during expiration (min size). All measurements, ratios, and indexes are reported in Table 2 and Table 3. There was a statistical difference (*p*-value < 0.0001) between max and min values for CVC (Figure 6). There was no statistical difference between the diameter in B-mode versus M-mode of the CVC measurements, CI CVC, and CVC/Ao (Figure 7).

### 3.3. Inguinal Variables

The inguinal vessels, FA and FV, showed no visual change in diameter attributable to the cardiac or respiratory cycle, so the vessels were only measured at their maximum size. Measurements and ratios are reported in Table 4 and Table 5.

### 3.4. Agreement for Respiratory Indexes and Vein-to-Artery Ratios

For the transrectal variables, repeatability and reproducibility were 93.9% and 94.4%, respectively. For the inguinal variables, repeatability and reproducibility were both 96.9%. Bland–Altman plots illustrate repeatability and reproducibility for transrectal (Figure 8) and inguinal (Figure 9) variables. For both techniques, repeatability plots (Figure 8A and Figure 9A) show tight clustering near the zero line and narrow limits of agreement, indicating high intra-operator consistency. Reproducibility plots (Figure 8B and Figure 9B) display a wider spread and more deviations, suggesting greater inter-operator variability.

## 4. Discussion

This is the first study to assess ultrasonographic evaluation of large blood vessels in healthy, sedated adult horses, providing baseline data on CI CVC and vein-to-artery ratios for future research on equine volume status and fluid responsiveness assessment.

The procedure was generally well tolerated. Some horses were stressed before or during the examination, highlighting the importance of sedation to ensure safety. In some horses, sedation wore off more quickly due to individual susceptibility. Rectal bleeding or pain after transrectal ultrasonography did not occur. However, pneumorectum, straining, and tension were common and could affect image acquisition. The authors believe that rectal instillation of a local anesthetic is not essential for the performance of transrectal ultrasonography but likely contributes to improved horse comfort and a decreased risk of complications.

Both ultrasonographers, one with moderate (OP1) and one with extensive (OP2) ultrasonographic experience, had no prior experience with inguinal or abdominal vessel imaging but found the technique manageable after a brief review of vessel anatomy and a few practice scans. This suggests that limited training may be sufficient for practitioners to acquire the skills needed for this procedure. The easiness of the scanning technique and the effect of training were also demonstrated by the short and decreasing examination times. Median examination times were namely only 3 (inguinal) and 4 min (transrectal) per horse and decreased from 12 to 16 min initially to as little as 2–3 min with increasing experience. For the examination to be potentially useful in emergency situations, it is important that it be rapid. Since transrectal palpation is a routine emergency procedure in many equine patients, and ultrasonographic equipment often is readily available, adding a transrectal or inguinal ultrasonographic protocol of only a couple of minutes appears feasible.

Vessel image quality was generally better transrectally than inguinally, though evaluation of CVC recordings was more challenging due to irregular changes in vessel diameter and difficulty in obtaining a transverse view. Inguinal recordings often had lower image quality but were easier to interpret and measure.

In our study, no diameter variations with cardiac and respiratory cycles were observed in inguinal vessels (FA, FV), while abdominal vessels exhibited marked changes with cardiac (Ao, CVC) and respiratory cycles (CVC). This contrasts with previous findings in dogs, foals, and calves, where no respiratory variation in CVC diameter is detected in the CVC at the paralumbar view [14,17,19,20]. This may reflect the deep, slow diaphragmatic movement in adult equines compared to the faster, shallower breathing of smaller or younger animals. Another possible explanation involves species-, breed-, and age-related differences in pleural pressure fluctuations during spontaneous respiration and their impact on CVC dimensions. For instance, the authors have observed marked CVC respiratory variation at the paralumbar site in brachycephalic dogs, likely due to the negative inspiratory pressure needed to overcome the inspiratory flow limitation compared to other dog breeds [31]. It could be hypothesized that CVC respiratory variations would be even more marked in non-sedated horses.

In our study, CVC respiratory variation allowed CI CVC calculation, ranging from mean values of 30–36% (diameter B-mode 30 ± 13%, M-mode 33 ± 12%; area 36 ± 15%) in healthy, sedated horses for both diameter and area, which is slightly higher than the 26% (diameter) reported in healthy, unsedated foals [18]. Differences in animal size, views, and methodologies limit direct comparison, as foals were examined using transabdominal and subxiphoid ultrasonography. In humans, several other factors are reported to influence the IVC diameter and CI IVC, including intrathoracic and intra-abdominal pressure, cardiac function, and local venous congestion or compression of the vessel [9], and could affect the reliability of CI CVC. For these reasons, severe asthma, intestinal tympanism, late pregnancy, or large abdominal masses could affect CI CVC in equine patients.

The CVC/Ao ratio in our study was 0.43 (diameter, mean) and 0.56 (area, median). These values are substantially lower than those in healthy foals (1.05 and 0.94 for diameter and area, respectively) or other species (calves: 0.71 (diameter) and 0.89 (area); dogs: reference range 0.93–1.32 (diameter); humans: reference range 0.8–1.2 (diameter)) [14,17,19,20,22,32,33]. In children, the IVC/Ao increases with age and is correlated with body size and surface area [34]. Positive correlation with age is also seen in calves and foals [19,20], and it is unclear why values in adult horses were lower compared to foals. One possible explanation is due to the larger extracellular fluid volume of foals [35] and differences in body or ultrasonographic positioning and respiratory cycle compared to adults [18,20]. Moreover, considering the large variability in body size in horses, this effect needs to be investigated in future studies, too. The CVC/Ao has the potential to identify hypovolemia [23] and identified volume depletion in dogs [23] that normalized after intravenous fluid therapy [15,23,24,25,26]. However, the CVC/Ao ratio may appear normal despite severe hypovolemia, as shown in a hemorrhagic shock model study in dogs [36], where volume depletion caused decreased CVC size as well as decreased cardiac output, which led to a reduced Ao size diameter. Just as for the CI CVC, inspiratory effort, cardiac conditions impeding venous return, and increased abdominal pressure [37] can also influence the CVC/Ao ratio.

The FV/FA diameter ratio correlates with central venous pressure (CVP) and also has the potential to identify hypovolemia in human patients [11]. Compared to human literature, the inguinal vessels were measured more distally in this present study [11]. Additionally, in horses, the difference between FV and FA is much larger, resulting in a higher ratio.

In this present study, both B-mode and M-mode were used to measure the diameter of the CVC and Ao, along with CVC/Ao and the CI CVC. We found no statistically significant differences between the two modes for any of the variables measured via transrectal ultrasonography. While M-mode requires more advanced equipment, B-mode is typically readily available for most equine clinicians. M-mode measurements can be influenced by cursor positioning, which can result in an underestimation or overestimation of vessel diameter [38]. However, B-mode measurements were more challenging due to difficulty in correctly placing the caliper, and the vessel diameter was not always aligned longitudinally with the ultrasonographic beam. This misalignment could introduce variability in the measurements. Taking all of the above into consideration, the authors preferred M-mode for CI CVC and CVC/Ao ratios. In this study, we measured only the short axis (height) of the vessels, unlike with the American College of Emergency Physicians’ [39] recommendation as applied by Barthelemy et al. (2023) [17], who used the mean of two perpendicular diameters in B-mode. Transrectal imaging made transverse B-mode views difficult, limiting accurate width assessment. Moreover, calipers were placed parallel to the ultrasound beam in B-mode to ensure consistency with M-mode and reduce measurement bias.

Repeatability and reproducibility were evaluated for the ultrasonographic technique (performed twice by two different operators), rather than for caliper placement during measurement of the different variables (which was performed by a single operator). This approach was chosen to specifically assess the effect of the operator on ultrasonographic techniques where vessel imaging is more likely to be influenced by both transducer positioning and the pressure applied during scanning, due to the close contact between the vessel and the ultrasound probe.

Repeatability and reproducibility were good for transrectal and excellent for inguinal variables. As expected with manual measurements, repeatability was higher than reproducibility for both variable types, since intra-observer measurements are generally more consistent than inter-observer measurements [40].

Limitations of this current study include a small number of horses for which variations in breed, gender, and age were not accounted for. Operators’ subjective interpretation of the images, changes in venous diameter, timing of breathing, transducer placement, the pressure applied to the vessels, and patient factors could have caused bias or amplified intra- and inter-rater variability. Ultrasound recordings were reviewed, and measurements were obtained by one of the operators who had performed the original examinations months earlier. As the operator was blinded, the risk of bias is considered minimal. The unknown effect of sedation on the ultrasonographic measurements and relative ratios and indexes is another limitation of this study. Xylazine is known to reduce cardiac output, heart rate, and respiratory rate and to increase the incidence of second-degree atrioventricular block, transient hypertension followed by hypotension, and increased systemic vascular resistance [41,42]. These effects could have a direct (via vasoconstriction) or indirect (by reducing cardiac output) impact on vessel size. Moreover, horses naturally exhibit a slow and deep breathing pattern, which is further slowed and unregularized by sedation, sometimes leading to periods of apnea. This could have resulted in inconsistent timing of the recorded measurements with inspiration, particularly affecting CI CVC. Even though the use of sedatives is common practice in equine emergency patients, future studies should investigate the effects of frequently used sedatives on these ultrasonographic measurements. In the meanwhile, one should be cautious to extrapolate the measurements of sedated horses to unsedated horses. Variations in the depth of sedation between horses, or within the same horse at different time points—particularly when sedation diminished during the examination—may have influenced the results. Furthermore, pooling of repeated measures could have induced type I error.

Future research should assess the differences between healthy and hypovolemic patients, ideally using clinical models of hemorrhage or hypovolemia, such as the study performed in healthy dogs after blood donation [22,43], or by following a protocol of volume depletion induced by furosemide [23] on mixed-effects models or repeated-measures analyses. Subsequent studies should focus on clinical cases of hypovolemia [24] and focus on identifying fluid responders [6,15] in order to propose cut-off indices that can guide fluid therapy in practical, clinical settings. Moreover, the influence of factors such as breed, body size, and common disease states such as intestinal tympanism and asthma needs to be explored in equine patients.

## 5. Conclusions

This pilot study demonstrates successful transrectal or inguinal ultrasonographic identification and measurement of the diameter and area of the CVC, Ao, FV, and FA in healthy, sedated, normovolemic horses, without encountering relevant complications. This technique allowed calculation of CI CVC, FV/FA ratio, and CVC/Ao ratio as described in other species and was shown to be repeatable, reproducible, and consistent across imaging modes (B- and M-mode). This study establishes baseline data for future research. As demonstrated in other species, this technique could have the potential to enhance emergency care and more accurately guide fluid therapy.

## Figures and Tables

**Figure 1 animals-15-02837-f001:**
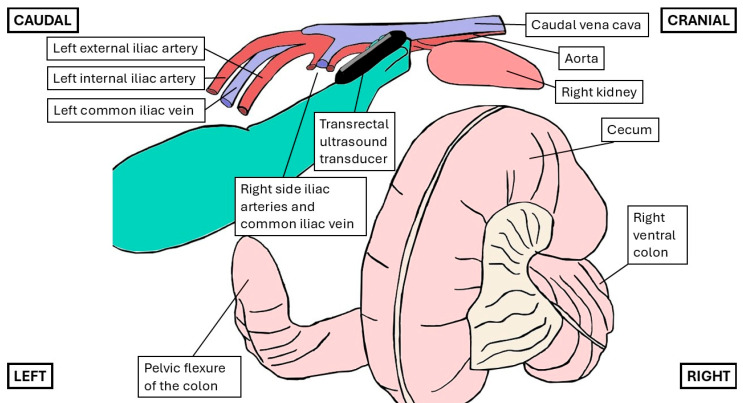
Diagram illustrating the positioning of the operator’s right hand during transrectal ultrasonographic examination. The hand was positioned dorsally in a sagittal or rightward scanning plane, approximately 50–60 cm cranial to the anus, and the ultrasound transducer was angled at approximately 45° toward the cranio-right abdominal quadrant, as shown in Figure 2A for the scan of the caudal vena cava and Figure 2B for the aorta. Note that several intra-abdominal organs, including the left kidney, duodenum, small intestines, and small colon, are not depicted in this representation. (Drawing based on The Glass Horse Project [28]).

**Figure 2 animals-15-02837-f002:**
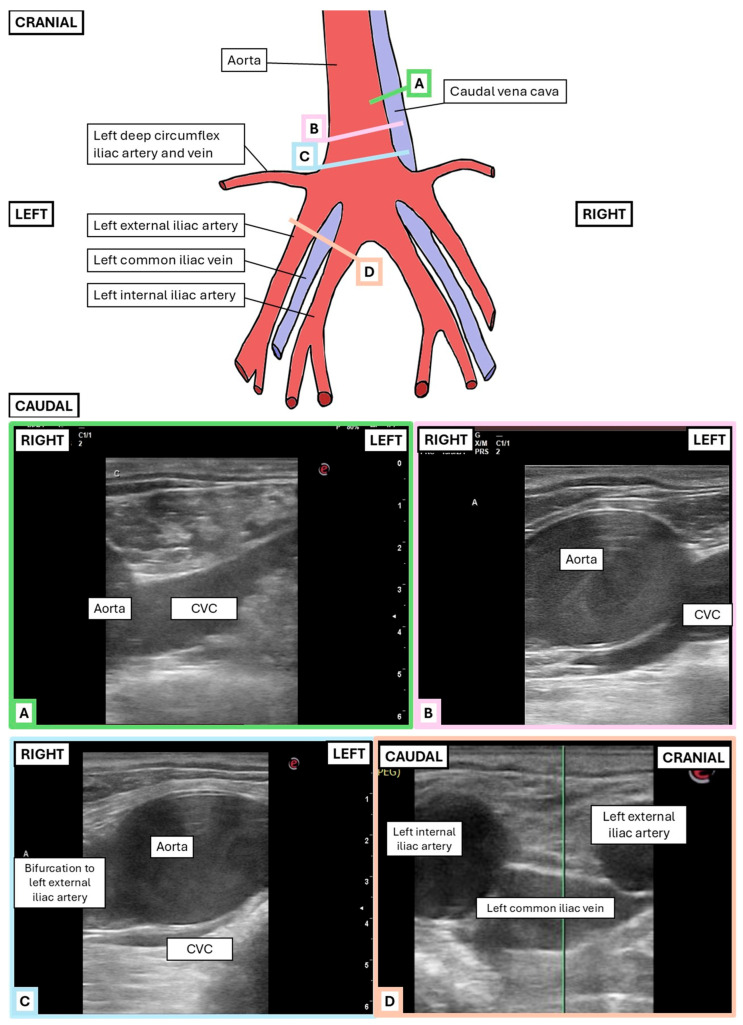
Illustration of the ventro-dorsal view of intra-abdominal vessels with corresponding B-mode ultrasonographic images acquired from different transducer positions, indicated by colored lines (positions (**A**–**D**)). (**A**) Transverse view of the intra-abdominal caudal vena cava (CVC). The image was obtained deep within the horse’s abdomen. At this level, the CVC is located laterally to the right of the aorta (Ao) and has a fusiform shape. This ultrasonographic view was used to measure the diameter and area of the CVC. (**B**) Transverse view of the intra-abdominal Ao. This image was taken slightly more caudal and was used to measure the diameter of the Ao. At this level, the CVC remains lateral to the right of the Ao, but its course becomes deeper, moving toward the midline. (**C**) This image, obtained slightly more caudally than the previous one, shows the CVC at the midline, positioned deeper than the Ao, along with the bifurcation of the left external iliac artery. (**D**) At the level where the abdominal Ao bifurcates (more caudally and laterally to the left compared to the other images), the left external and internal iliac arteries are visible. Between them, slightly deeper, lies the left common iliac vein. (Drawing based on Denoix, J.-M. (2019) [29]).

**Figure 3 animals-15-02837-f003:**
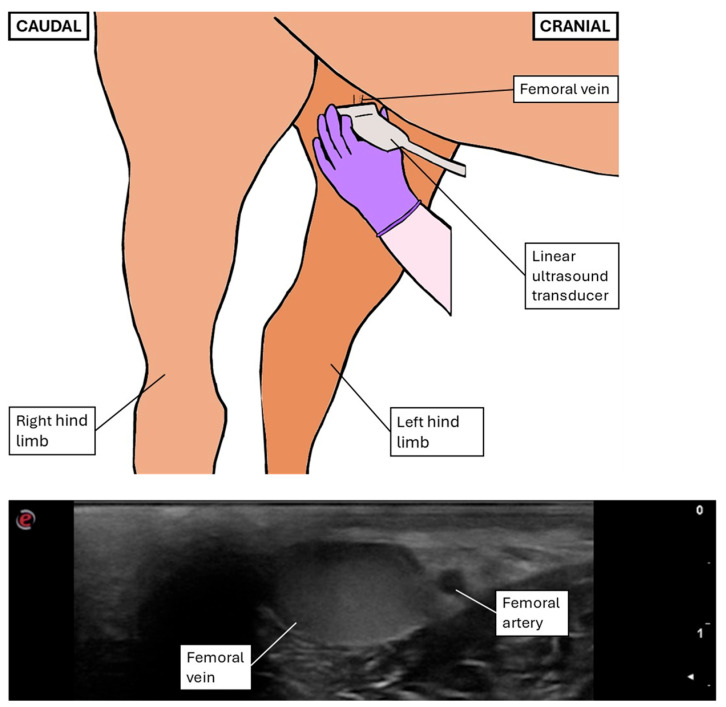
Diagram illustrating the positioning of the operator’s left hand during transcutaneous inguinal ultrasonographic examination and corresponding B-mode ultrasonographic image obtained.

**Figure 4 animals-15-02837-f004:**
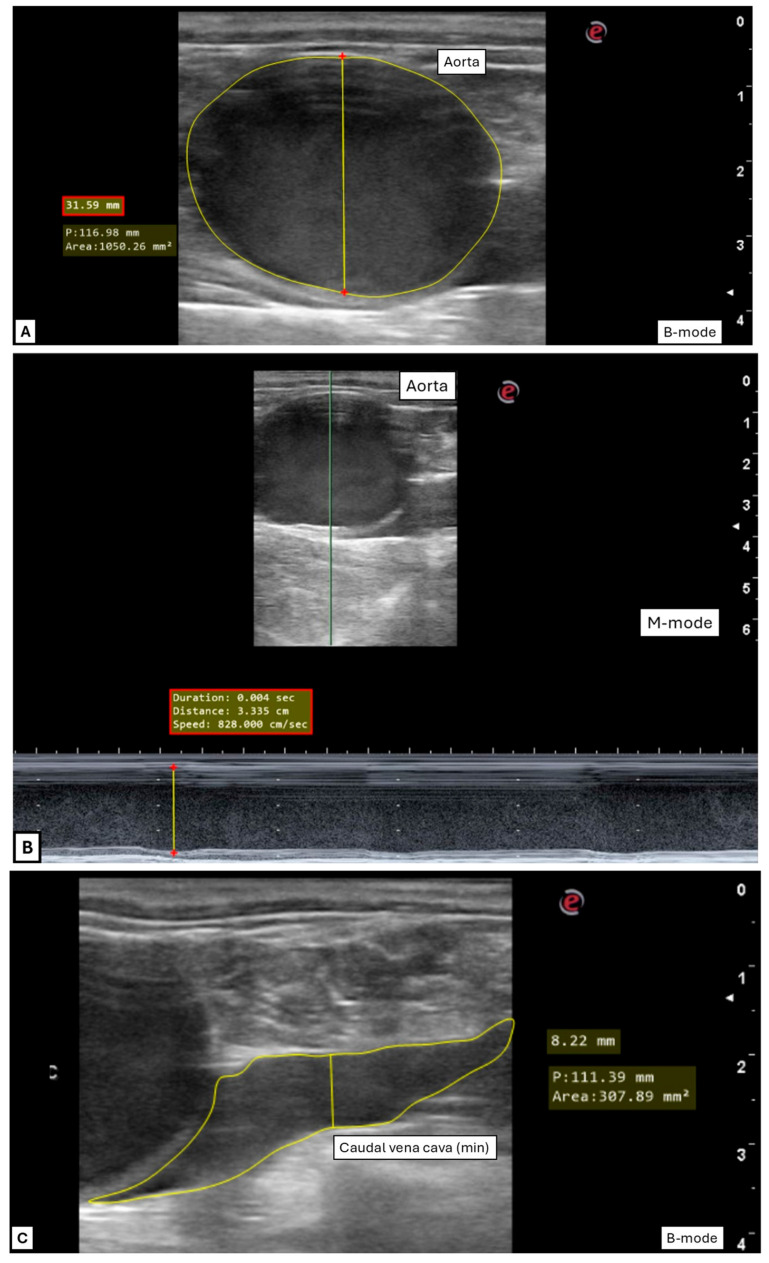
Ultrasonographic images for vessel measurements. (**A**) Transrectal B-mode: aorta in systole, diameter, and area; (**B**) transrectal M-mode: aorta in systole, diameter; (**C**) transrectal B-mode: caudal vena cava (CVC) minimum diameter and area; (**D**) transrectal B-mode: CVC maximum diameter and area; (**E**) transrectal M-mode: CVC minimum and maximum diameter; (**F**) transcutaneous inguinal B-mode: femoral artery and vein. Images are taken from the same horse except for FA and FV (**F**).

**Figure 5 animals-15-02837-f005:**
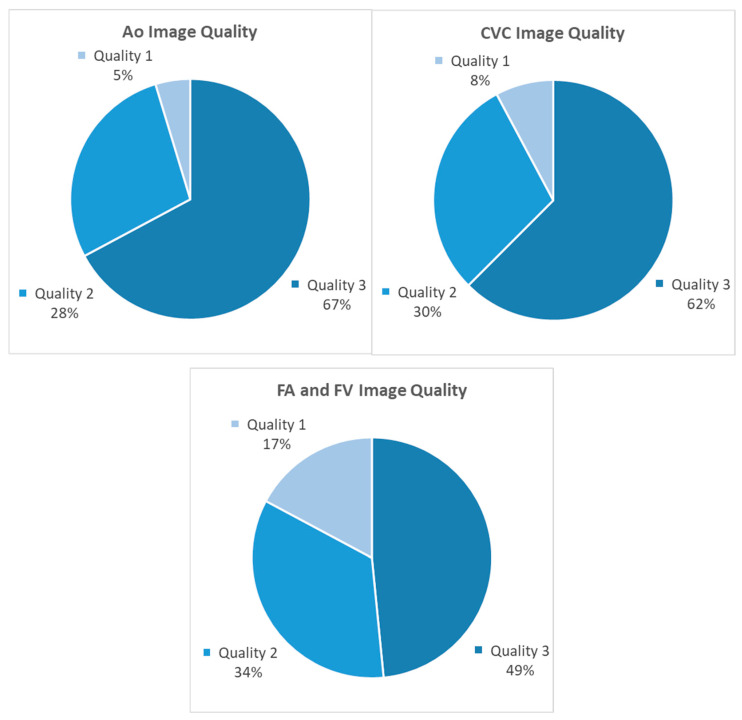
Classification of ultrasonographic image quality for intra-abdominal vessels (aorta, caudal vena cava) and inguinal vessels (femoral artery and vein). Image quality was graded on a scale from 1 to 3: (1) artifacts prevent full visualization or precise delineation of one or more vessels; (2) artifacts are present but only partially affect vessel delineation (e.g., obscuring one edge); (3) high-quality image with no artifacts. Abbreviations: Ao = aorta; CVC = caudal vena cava; FA = femoral artery; FV = femoral vein.

**Figure 6 animals-15-02837-f006:**
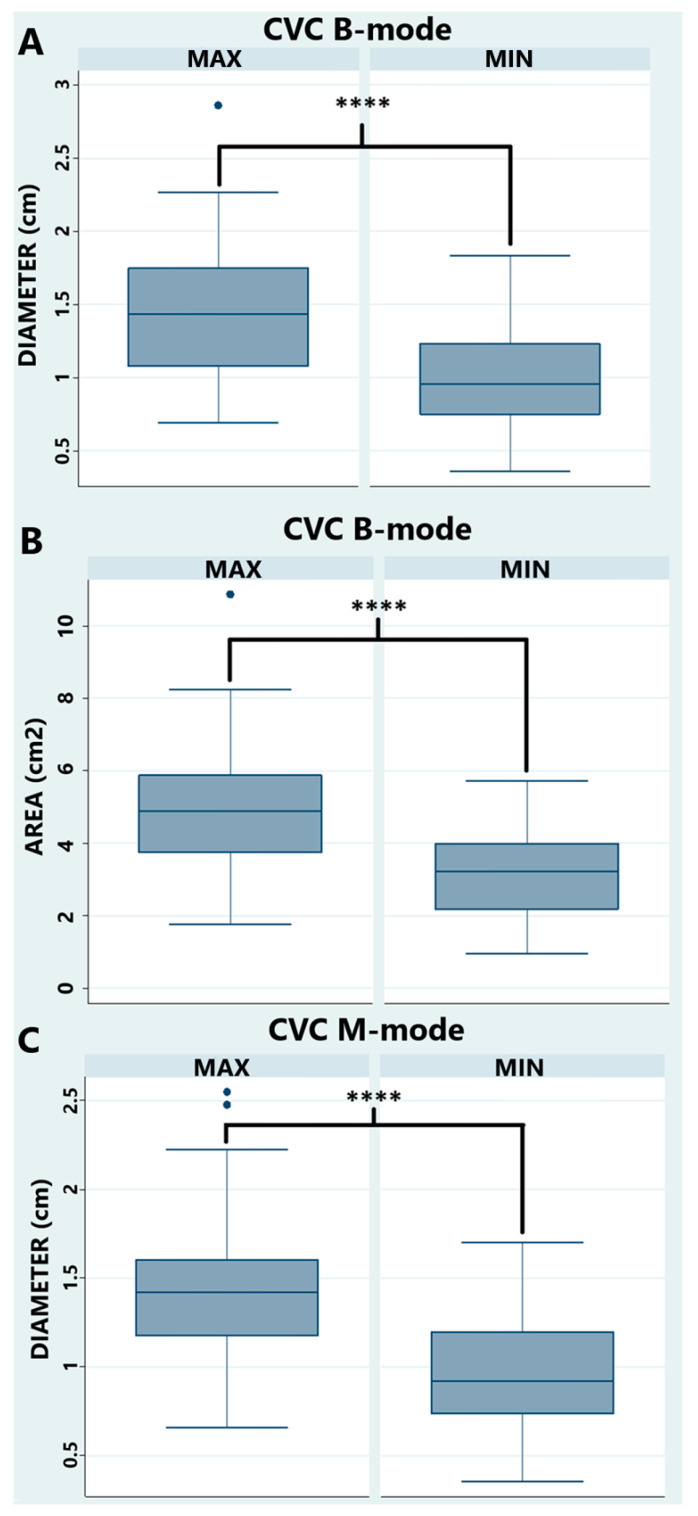
Comparison between maximum and minimum value of caudal vena cava in B-mode for the diameter (**A**) and area (**B**); in M-mode for the diameter (**C**). Abbreviations: CVC = caudal vena cava; CVC min/max = caudal vena cava minimum/maximum diameter or area. **** *p* < 0.0001.

**Figure 7 animals-15-02837-f007:**
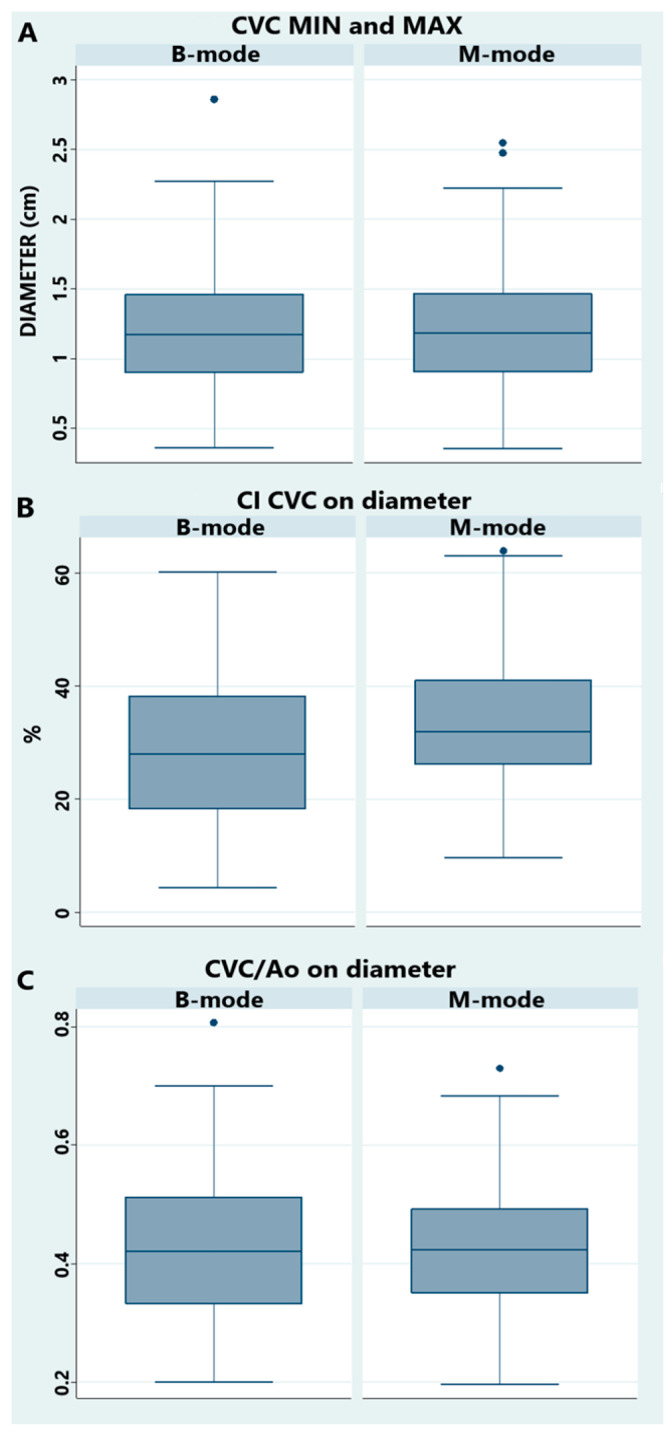
Comparison between B- and M-mode for caudal vena cava (CVC) diameter maximum and minimum value (**A**), CVC respiratory collapsibility on diameter (**B**), and CVC-to-aorta ratio on diameter (**C**). *p*-values were 0.98 (**A**), 0.11 (**B**), and 0.79 (**C**). Abbreviations: Ao = intrabdominal aorta, CVC = caudal vena cava, CI CVC = respiratory collapsibility index of the caudal vena cava, CVC/Ao = caudal vena cava-to-aorta ratio.

**Figure 8 animals-15-02837-f008:**
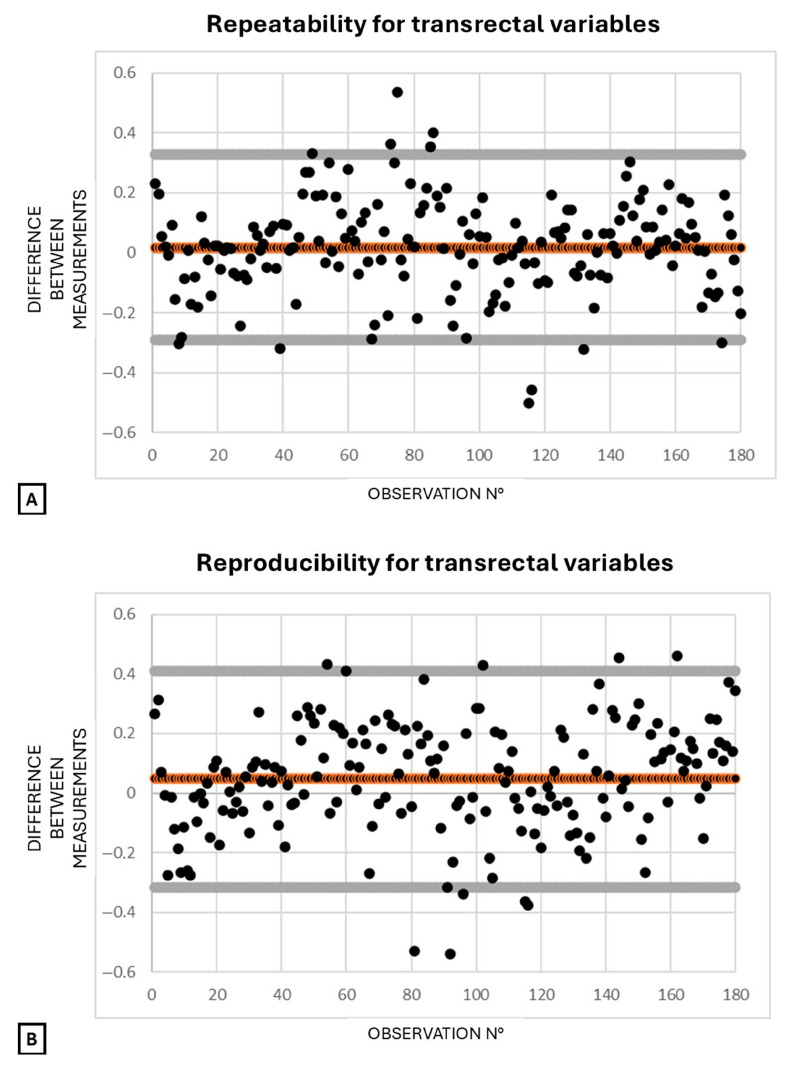
Bland–Altman plots for transrectal variables (CI CVC and CVC/Ao from B-mode diameter, B-mode area, and M-mode diameter), showing the difference between two measurements by the same operator (repeatability, (**A**)) or two different operators (reproducibility, (**B**)), plotted against observation number. Each black dot represents one observation; orange-bordered dots indicate the mean difference. Dots near the center suggest good agreement, while those farther away indicate greater disagreement. Gray lines represent the 95% limits of agreement. Abbreviations: Ao = aorta; CVC = caudal vena cava; CI CVC = respiratory collapsibility index of the caudal vena cava; CVC/Ao = caudal vena cava-to-aorta ratio.

**Figure 9 animals-15-02837-f009:**
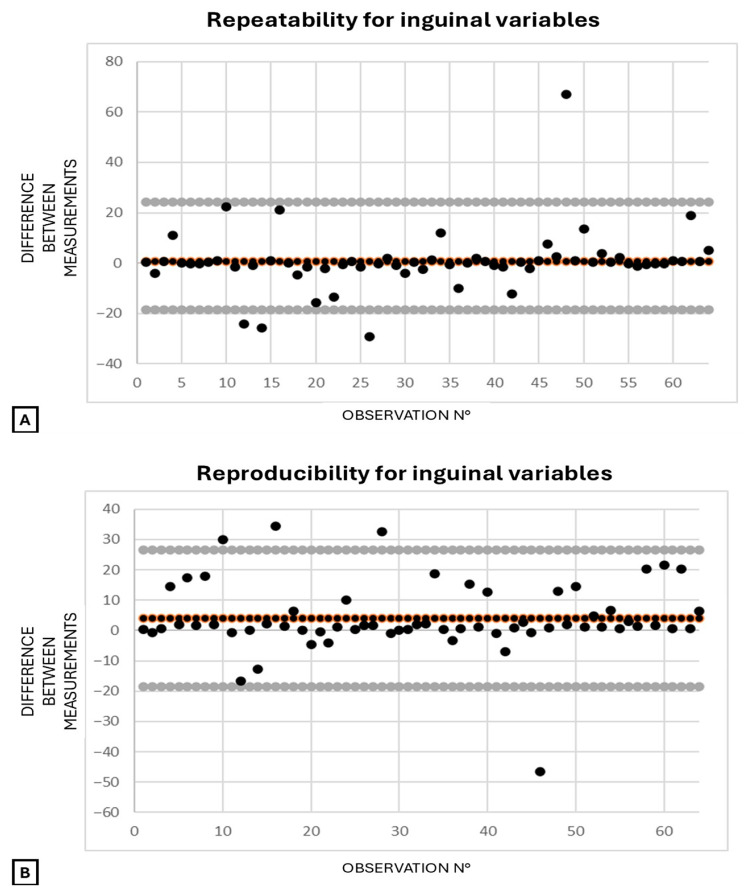
Bland–Altman plots for inguinal variables (FV/FA from B-mode diameter and area), showing the difference between two measurements by the same operator (repeatability, (**A**)) or two different operators (reproducibility, (**B**)), plotted against observation number. Each black dot represents one observation; orange-bordered dots indicate the mean difference. Dots near the center suggest good agreement, while those farther away indicate greater disagreement. Gray lines represent the 95% limits of agreement. Abbreviations: FA = femoral artery; FV = femoral vein; FV/FA = femoral vein-to-artery ratio.

**Table 1 animals-15-02837-t001:** All ultrasonographic variables that were measured and calculated in B-mode and M-mode during transrectal and inguinal examinations.

	B-Mode	M-Mode
Transrectal examination		
Measured variables		
Ao	Diam (cm)	Diam (cm)
	Area (cm^2^)	
CVC min	Diam (cm)	Diam (cm)
	Area (cm^2^)	
CVC max	Diam (cm)	Diam (cm)
	Area (cm^2^)	
Calculated variables		
CI CVC	=(CVC diam max -CVC diam min)/(CVC diam max)	=(CVC diam max -CVC diam min)/(CVC diam max)
	= (CVC area max -CVC area min)/(CVC area max)	
CVC/Ao	=(CVC diam max/Ao diam)	=(CVC diam max/Ao diam)
	=(CVC area max/Ao area)	
Inguinal examination		
Measured variables		
FA	Diam (cm)	n/a
	Area (cm^2^)	
FV	Diam (cm)	n/a
	Area (cm^2^)	
Calculated variables		
FV/FA	=(FV diam max/FA diam)	n/a
	=(FV area max/FA area)	

Abbreviations: Ao = aorta; CVC = caudal vena cava; CVC min/max = caudal vena cava minimum/maximum diameter or area; CI CVC = respiratory collapsibility index of the caudal vena cava; CVC/Ao = caudal vena cava-to-aorta ratio; Diam = diameter; FA = femoral artery; FV = femoral vein; FV/FA = femoral vein-to-artery ratio.

**Table 2 animals-15-02837-t002:** Transrectal ultrasonographic measurements of the aorta and caudal vena cava in B-mode and M-mode.

	B-Mode		M-Mode	
Vessel	Mean ± SD	95% Confidence Interval (Lower-Upper)	Mean ± SD	95% CI (Lower-Upper)
Ao diam (cm)	3.3 ± 0.3	(3.23–3.40)	3.3 ± 0.3	(3.25–3.42)
Ao area (cm^2^)	8.7 ± 1.7	(8.27–9.14)	n/a	n/a
CVC diam min (cm)	1.0 ± 0.3	(0.91–1.07)	1.0 ± 0.3	(0.88–1.04)
CVC diam max (cm)	1.4 ± 0.4	(1.32–1.53)	1.4 ± 0.4	(1.34–1.54)
CVC area min (cm^2^)	3.2 ± 1.2	(2.86–3.46)	n/a	n/a
CVC area max (cm^2^)	5.0 ± 1.7	(4.56–5.43)	n/a	n/a

Abbreviations: Ao = aorta; CI = confidence interval; CVC = caudal vena cava; CVC min/max = caudal vena cava minimum/maximum diameter or area; Diam = diameter; SD = standard deviation.

**Table 3 animals-15-02837-t003:** Respiratory collapsibility index of the caudal vena cava and caudal vena cava-to-aorta ratio measured through transrectal ultrasonography in B-mode and M-mode.

	B-Mode			M-Mode		
	Mean ± SD	Median (IQR)	95% CI (Lower-Upper)	Mean ± SD	Median (IQR)	95% CI (Lower-Upper)
CI CVC on diam	0.30 ± 0.13	n/a	(0.27–0.33)	0.33 ± 0.12	n/a	(0.30–0.36)
CI CVC on area	0.36 ± 0.15	n/a	(0.32–0.40)	n/a	n/a	n/a
CVC/Ao diam	0.43 ± 0.15	n/a	(0.40–0.46)	0.43 ± 0.11	n/a	(0.40–0.46)
CVC/Ao area *	n/a	0.56 (0.53–0.64)	n/a	n/a	n/a	n/a

Abbreviations: Ao = aorta; CVC = caudal vena cava; CI = confidence interval; CI CVC = respiratory collapsibility index of the caudal vena cava; CVC/Ao = caudal vena cava-to-aorta ratio; Diam = diameter; IQR = interquartile range; SD = standard deviation. CI CVC was calculated as follows: ((CVC max value-CVC min value)/CVC max value). CVC/Ao was calculated as follows: CVC max value /Ao value. * Data with non-normal distribution.

**Table 4 animals-15-02837-t004:** Measurements of the femoral artery and vein measured with transcutaneous inguinal ultrasonography in B-mode.

Vessel	Median (IQR)
FA diam (cm) *	0.2 (0.22–0.25)
FA area (cm^2^) *	0.04 (0.04–0.05)
FV diam (cm) *	0.8 (0.82–0.97)
FV area (cm^2^) *	1.0 (1.08–1.46)

Abbreviations: Diam = diameter; FA = femoral artery; FV = femoral vein; FV/FA = femoral vein-to-artery ratio; IQR = interquartile range. * Data with non-normal distribution.

**Table 5 animals-15-02837-t005:** Measurements of femoral vein-to-artery ratio in B-mode.

	Mean ± SD	Median (IQR)	95% CI (Lower-Upper)
FV/FA diam	3.80 ± 1.02	n/a	(3.54–4.05)
FV/FA area *	n/a	26.74 (25.23–32.40)	n/a

Abbreviations: CI = confidence interval; Diam = diameter; FA = femoral artery; FV = femoral vein; FV/FA = femoral vein-to-artery ratio; IQR = interquartile range; SD = standard deviation. FV/ FA was calculated for the values of diameter and area as follows: FV value /FA value. * Data with non-normal distribution.

## Data Availability

The data that support the findings of this study are available on request from the corresponding author. The data are not publicly available due to privacy or ethical restrictions.

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
