# Peer review of "Ultrasonographic Assessment of Caudal Vena Cava Collapsibility Index, Caudal Vena Cava-to-Aorta, and Femoral Vein-to-Artery Ratios in Healthy Sedated Adult Horses"

_animals, 2025, doi:10.3390/ani15192837_

Round 1

Reviewer 1 Report

Comments and Suggestions for Authors

This pilot study shows the potential validity of this new technique and on the whole is well written. Please correct the manuscript where the flollowing error message is found "Error! Reference source not found". Also, please see comments below:

Line 43: There are various types of lactic acidosis (A, B, C, D) but is venous B-lactate a traditional measurement technique. L-lactate is usually measured in blood.

Line 95: When was PCV and L-lactate measured in relation to the ultrasound exam?

Line 99: How much time was there between xylazine administration and beginning of the ultrasound exam? Was it the time the same for all horses? What clinical signs did the horse have to exhibit before the scan commenced? Did any horse require additional doses of xylazine? Were the horses observed for any adverse effects after the scans had finished, e.g. colic, and for how long after the scans?

Line 129: How was each horse restrained for the rectal exam? Were stocks used?

Line 132-134: How long after the instillation of local anesthetic did the scan begin? Was this the same for lidocaine and mepivacaine? Was the time the same for every horse?

Line 141: Both operators scanned the horse in the same exam? Was the order of who scanned first the same or randomised. Would separate exams for each operator have led to more consistency, as sedation and local anesthetic effect would have potentially been at the same level for each scan, instead of the second operator following the first with sedation and local anesthetic effects decreasing?

Line 167: How was each horse restrained for the inguinal exam? Were stocks used?

Table 1: This table is probably not necessary as the variables to be measured are explained in the text as are the formulas. Or if the authors prefer, the text should be removed with readers directed to Table 1.

Line 222: Formula for CVC/Ao is missing.

Line 229: Formula for FV/FA is missing.

Line 231: Was an initial power analysis carried out to determine the minimum number of horses required to produce meaningful data. Please provide calculation if this was done. If a power analysis was not carried out, then please explain why.

Line 387: While the exams were rapid, please comment on the time taken for sedation and local anesthetic instillation which would add time to the overall scan. Do the authors feel the local anesthetic instillation is essential for this procedure?

Line 426: Could sedation of also been a factor?

Line 469: Some bias may have been reduced by the images being reviewed blindly. However, what about the bias of only one operator looking at the images? For example, positioning of the measurement calipers on the image is very subjective and could have created bias with only one person. This could also have enabled the authors to assess how easy or hard it was to obtain the measurements from the images. Why did the authors decide not to use more than one person to view the images?

Author Response

We sincerely thank Reviewer 1 for their thorough review and valuable feedback. We have addressed each comment below and made the corresponding revisions to the manuscript.

Please correct the manuscript where the flollowing error message is found "Error! Reference source not found".

This has been corrected.

Line 43: There are various types of lactic acidosis (A, B, C, D) but is venous B-lactate a traditional measurement technique. L-lactate is usually measured in blood.

This has been corrected.

Line 95: When was PCV and L-lactate measured in relation to the ultrasound exam?

PCV and L-lactate were measured before sedation and ultrasound examination. This was clarified in the manuscript.

Line 99: How much time was there between xylazine administration and beginning of the ultrasound exam? Was it the time the same for all horses? What clinical signs did the horse have to exhibit before the scan commenced? Did any horse require additional doses of xylazine? Were the horses observed for any adverse effects after the scans had finished, e.g. colic, and for how long after the scans?

No specific timing between xylazine administration and beginning of the ultrasound exam, which was simply started when the horse showed tireness and sedated behaviour, which variated for different horses depending on the resulting grade of sedation: mild when the horse exhibited slight ptosis and a reduced, delayed response to environmental stimuli; and as moderate when characterized by marked ptosis, absence of response to environmental stimuli, and ataxia. Generally speaking, examinations were commenced around 5 minutes after the administration of the sedative.
No horse received addictional doses of sedatives. Horses were observed by the owners at home or at the clinic, with no horses having any addictional discomfort after examinations - the horses tended to be almost fully awake when the examinations were finished.

Information was improved accordinly in the manuscript: Line 109-110, Line 289-290.

Line 129: How was each horse restrained for the rectal exam? Were stocks used?

Horses were restrained in a stock when examined at the clinic, but no additional restrain if examined at the horses' stable. This information was improved in the manuscript line 111-113.

Line 132-134: How long after the instillation of local anesthetic did the scan begin? Was this the same for lidocaine and mepivacaine? Was the time the same for every horse?

The local anaesthetic was instilled after sedation, and the ultrasound examinations were started once the horses showed signs of sedation.

Line 141: Both operators scanned the horse in the same exam? Was the order of who scanned first the same or randomised. Would separate exams for each operator have led to more consistency, as sedation and local anesthetic effect would have potentially been at the same level for each scan, instead of the second operator following the first with sedation and local anesthetic effects decreasing?

Yes, all examinations were done one after the other under the same occasion. The order was consistent: operator 1 scanned first, then operator 2. Separate exams for each operator could have possibly led to more consistent ground condition (same grade of sedation during the different exams for the different operators); on the other hand, this would have complicated the experimental setting of this study which was conducted in a private equine hospital with limited fundings, making it more difficult to practically conduct it by asking the free willing owners to investigate their horses twice. So decision was made to perform all the exams under the same occasion. This is mentioned as a limitation in the discussion section (line 507-509).

Line 167: How was each horse restrained for the inguinal exam? Were stocks used?

Same as for the transrectal examination.

Table 1: This table is probably not necessary as the variables to be measured are explained in the text as are the formulas. Or if the authors prefer, the text should be removed with readers directed to Table 1.

The table presents a more systematic overview of what was measured and for each ultrasonographic technique. For instance, it clarifies that area measurements were not performed using M-mode imaging, and that M-mode was not used for inguinal ultrasonography. Although we had previously attempted to convey this information in the text, we recognized that presenting it in a table would make it clearer and more accessible to the reader.

Line 222: Formula for CVC/Ao is missing.

Line 229: Formula for FV/FA is missing.

Thank you for noticing this. The missing elements have been reinserted in the revised manuscript.

Line 231: Was an initial power analysis carried out to determine the minimum number of horses required to produce meaningful data. Please provide calculation if this was done. If a power analysis was not carried out, then please explain why.

No, an initial power analysis was not carried out.  Our study is a pilot study aiming at collecting preliminary data. The aim of the study was not to find differences between groups of horses (for ex healthy vs sick). Instead convenience sample of more than 15 horses was chosen. This information was added in the text line 89-91.

Line 387: While the exams were rapid, please comment on the time taken for sedation and local anesthetic instillation which would add time to the overall scan. Do the authors feel the local anesthetic instillation is essential for this procedure?

Fortunately, sedation and instillation of local anaesthetic did not add so much time to the overall scan. Sedation is something is rutinely done in clinical settings before invasive investigations such as transrectal palpation. Intra-rectal local anaesthetic instillation is not rutely done, is definitely not essential for the procedure but an add-on for ensuring extra-safety since this type of transrectal ultrasonography is indeed very deep into the horse. 

Line 426: Could sedation of also been a factor?

Yes, and this is presented as a limitation of the study (line 495-509).

Line 469: Some bias may have been reduced by the images being reviewed blindly. However, what about the bias of only one operator looking at the images? For example, positioning of the measurement calipers on the image is very subjective and could have created bias with only one person. This could also have enabled the authors to assess how easy or hard it was to obtain the measurements from the images. Why did the authors decide not to use more than one person to view the images?

This is correct, there could be an influence of caliper positioning on the obtained measurements, potentially leading to some degree of inter-rater variability. Since this is a pilot study evaluating the feasibility of ultrasonography of large blood vessels, we decided to focus on evaluating the intra and inter-rater variability of our ultrasonographic technique rather than on the inter-rater variability in caliper positioning. I added a commentary on this in discussion (line 477-483).

Reviewer 2 Report

Comments and Suggestions for Authors

This study demonstrated for the first time the feasibility of obtaining intrarectal measurements of caudal vena cava and aorta and inguinal transcutaneous measurements of femoral artery and vein in horses.

Measurements of major vessels are routinely used in other species and give valuable information concerning the patient fluid status; this study opens the door for future research on using these measurements to address the hydration status in horses.

The study is well structured and well-written; I strongly appreciated the practical approach to the subject and I found the images really helpful in the comprehension of the ultrasonographic technique.

In my opinion, a main limitation is represented by the fact that, despite images were obtained by two different operators, ultrasonographic measurements were performed by the same operator; consequently, this does not tell us how measurements would vary between different operators. To check that, multiple people should measure the same images independently.

Specific comments:

Figure 2: I think it would be easier to the reader to visualize the images in the right order: A,B,C,D instead of C,D,A,B.

Also, for figures A,B and C, does the left of the image correspond to the left of the horse? Please specify it

Figure 4B: “Transrectal M-mode: aorta in systole, diameter”. Please specify how you determined you actually obtained the image during systole. Please add this information (measurements of aorta obtained during systole) to materials and methods

Lines 216: in the paragraph “general variables” it is said that areas were measured following the internal lining of the vessels ‘wall. Was the area of the aorta calculated from the diameter or obtained following the internal lining of the vessel?

Lines 220-223: these data are already reported in table 1

Lines 228-229: the phrase is incomplete

Figure 5; lines 298-300: Image quality grading should be described in materials and methods

Table 2: Please mark the significantly different results

Line 336: Please specify to which table you are referring to

Line 347-349: I find the statement “repeatability and reproducibility were 93,9% and 94,4%..” too general. I suggest reporting repeatability and reproducibility limits and also specifying whether these values are considered excellent, good, moderate etc.  

Author Response

We sincerely thank Reviewer 2 for their thorough review and valuable feedback. We have addressed each comment below and made the corresponding revisions to the manuscript.

In my opinion, a main limitation is represented by the fact that, despite images were obtained by two different operators, ultrasonographic measurements were performed by the same operator; consequently, this does not tell us how measurements would vary between different operators. To check that, multiple people should measure the same images independently.

This is correct, there could be an influence of caliper positioning on the obtained measurements, potentially leading to some degree of inter-rater variability. Since this is a pilot study evaluating the feasibility of ultrasonography of large blood vessels, we decided to focus on evaluating the intra and inter-rater variability of our ultrasonographic technique rather than on the inter-rater variability in caliper positioning. I added a commentary on this in discussion (line 477-483).

Figure 2: I think it would be easier to the reader to visualize the images in the right order: A,B,C,D instead of C,D,A,B.

This was corrected.

Also, for figures A,B and C, does the left of the image correspond to the left of the horse? Please specify it

In those figures, left is left and right is right. Information in the images was improved accordinly.

Figure 4B: “Transrectal M-mode: aorta in systole, diameter”. Please specify how you determined you actually obtained the image during systole. Please add this information (measurements of aorta obtained during systole) to materials and methods

The Ao was measured when the vessel presented a clear pulsation and the diameter was at the largest (systole). This information was improved in lines 232-233.

Lines 216: in the paragraph “general variables” it is said that areas were measured following the internal lining of the vessels ‘wall. Was the area of the aorta calculated from the diameter or obtained following the internal lining of the vessel?

The area of the aorta was calculated from the diameter (line 233).

Lines 220-223: these data are already reported in table 1

The table presents a more systematic overview of what was measured and for each ultrasonographic technique. For instance, it clarifies that area measurements were not performed using M-mode imaging, and that M-mode was not used for inguinal ultrasonography. Although we had previously attempted to convey this information in the text, we recognized that presenting it in a table would make it clearer and more accessible to the reader.

Lines 228-229: the phrase is incomplete

This was corrected.

Figure 5; lines 298-300: Image quality grading should be described in materials and methods

This was described in material and methods, line 200-204. Please let us know if any further adjustments should be made.

Table 2: Please mark the significantly different results

Significantly different results are reported in figures 6 and 7.

Line 336: Please specify to which table you are referring to

This was corrected.

Line 347-349: I find the statement “repeatability and reproducibility were 93,9% and 94,4%..” too general. I suggest reporting repeatability and reproducibility limits and also specifying whether these values are considered excellent, good, moderate etc.  

Disagreement limit was consider 5%, information was added line 259. Values of repeatability and reproducibility were judged therby good for transrectal and excellent for inguinal variables (this is reported in discussion, line 484-485). Please let us know if any further information should be added in the result section.

Reviewer 3 Report

Comments and Suggestions for Authors

The paper is a prospective study evaluating feasibility and operator agreement of transrectal and inguinal US to quantify caudal vena cava collapsibility and vein to artery ratios in healthy sedated adult horses as base work for volume status assessment.

The abstract is well written, however please consider that excessive numbers reduce readability. Abstracts should highlight the most important findings rather than provide all mean, median, IQR values.

Justification of sample size is lacking, and no power calculation or rationale for 17 horses is provided, please do so (even if convenience sample, please mention it). 

Regarding methodology, the pooling of repeated measures and treating them like independent values, while they are not, might increase type 1 errors. I would at least include this in the limitations and mention it in the context of future work (mixed-effects models or repeated-measures analyses would increase robustness of findings).

It would be also important to mention that the analyzed indices are baseline descriptors, but without validation against true volume status (blood loss models, fluid therapy responses) cannot be used for decision making (also important for future studies).

Please check lines 208, 209, 220, 222, 227, 228, 307, 335, probably a reference software error.

Overall I enjoyed reading your paper and feel like the study lays groundwork but requires disease-model validation before it can change practice.

Author Response

We sincerely thank Reviewer 3 for their thorough review and valuable feedback. We have addressed each comment below and made the corresponding revisions to the manuscript.

The abstract is well written, however please consider that excessive numbers reduce readability. Abstracts should highlight the most important findings rather than provide all mean, median, IQR values.

Justification of sample size is lacking, and no power calculation or rationale for 17 horses is provided, please do so (even if convenience sample, please mention it). 

Initial power analysis was not carried out. Our study is a pilot study aiming at collecting preliminary data. The aim of the study was not to find differences between groups of horse (for ex healthy vs sick). Instead convenience sample of more than 15 horses was chosen. This information was added in the text line 89-91.

Regarding methodology, the pooling of repeated measures and treating them like independent values, while they are not, might increase type 1 errors. I would at least include this in the limitations and mention it in the context of future work (mixed-effects models or repeated-measures analyses would increase robustness of findings).

This information was improved in the limitation section as suggested (line 510; 511-515).

It would be also important to mention that the analyzed indices are baseline descriptors, but without validation against true volume status (blood loss models, fluid therapy responses) cannot be used for decision making (also important for future studies).

This was improved in Line 515-517 and 526.

Please check lines 208, 209, 220, 222, 227, 228, 307, 335, probably a reference software error.

This was corrected.

Reviewer 4 Report

Comments and Suggestions for Authors

I thank the authors for submitting their manuscript. The research addresses an innovative and significant topic for both the journal and clinical practice, promising a substantial impact on enriching current knowledge. This work presents important advancements in the research field and highlights a topic of interest to readers. The proposed methodology includes essential elements of the research; however, certain aspects of the manuscript need clarification. I have included some comments that I hope will be useful to the authors.

General comments

L19-22: I recommend that the authors revise the aims, as the current wording is unclear.

L46: In other species, more precise indicators are available to guide fluid therapy responses, such as the plethysmographic variability index (PVi). This comparative aspect should also be discussed.

10.3390/vetsci11090396

10.1111/vec.12728

10.1111/j.1399-6576.2011.02435.x

L78: Please include your hypothesis.

L81, 85, 112, and 113: missing data in the paragraph wording.

L85: Can you clarify how the sample size was calculated?

L103: Was a sedation scale used on the horses? If yes, please specify which one.

L126: Which scale was used? Please clarify.

L130: When citing equipment, materials, drugs, reagents, software, programs, or statistical packages, include the name, brand, model, laboratory, concentration, software or program version, and country of origin in parentheses, as appropriate. Apply this change throughout the manuscript.

L220, 222, 228, 307, 336: review, add references, and remove the phrase, Error! Reference source not found.

L260: What scale are you referring to? Please clarify.

L378: In how many cases did this happen?

L370: Was interobserver reliability evaluated? What are the sensitivity and specificity of the proposed method?

L376: Please explain why the xylazine doses in four horses had a shorter biological half-life.

L492: I suggest that the authors revise the conclusions to align with the aims described in L72-78.

L498-504: The points related to these aspects are missing.

Author Response

We sincerely thank Reviewer 4 for their thorough review and valuable feedback. We have addressed each comment below and made the corresponding revisions to the manuscript.

L19-22: I recommend that the authors revise the aims, as the current wording is unclear.

The aims have been rephrased for improved clarity. Please don’t hesitate to indicate if further or more specific adjustments are desired.

L46: In other species, more precise indicators are available to guide fluid therapy responses, such as the plethysmographic variability index (PVi). This comparative aspect should also be discussed.

10.3390/vetsci11090396

10.1111/vec.12728

10.1111/j.1399-6576.2011.02435.x

We thank the reviewer for the suggestion and the valuable insight. However, we respectfully chose not to implement this change. The Pleth Variability Index (PVi) has shown potential primarily in settings involving controlled breathing and minimal motion, such as during mechanical ventilation under general anesthesia. These conditions differ significantly from those in our study, which was conducted in standing horses under mild to moderate sedation. Furthermore, one of the studies referenced by the reviewer (Soubrier et al., 2011; DOI: 10.1111/j.1399-6576.2011.02435.x) concluded that PVi did not reliably predict fluid responsiveness in humans, even under controlled conditions. Given these limitations, and in the interest of maintaining focus and clarity in a manuscript that is already dense with content and references, we have opted not to include a discussion of PVi at this time.

L78: Please include your hypothesis.

An hypothesis was now included (line 81-82).

L81, 85, 112, and 113: missing data in the paragraph wording.

All missing data were added.

L85: Can you clarify how the sample size was calculated?

Power analysis was not carried out since the exploratory nature of this pilot study, which was carried in order to collect some preliminary data and not in order to find differences between groups of horses (for example, healthy vs sick). So a convenience sample of more than 15 horses was chosen instead. The information was added in the text line 89-91.

L103: Was a sedation scale used on the horses? If yes, please specify which one.

No specific sedation scale was used. Definition of mild and moderate sedation was described in Line 109-112.

L126: Which scale was used? Please clarify.

No specific sedation scale was used. Definition of mild and moderate sedation was described in Line 109-112.

L130: When citing equipment, materials, drugs, reagents, software, programs, or statistical packages, include the name, brand, model, laboratory, concentration, software or program version, and country of origin in parentheses, as appropriate. Apply this change throughout the manuscript.

This has been improved throughout the manuscript.

L220, 222, 228, 307, 336: review, add references, and remove the phrase, Error! Reference source not found.

This was corrected.

L260: What scale are you referring to? Please clarify.

No specific sedation scale was used. Definition of mild and moderate sedation was described in Line 109-112.

L378: In how many cases did this happen?

This is reported in results in lines 296-298.

L370: Was interobserver reliability evaluated? What are the sensitivity and specificity of the proposed method?

Inter-rater variability, determined by calculations of repeatability, was evaluated according to the study by Watson and Petrie (https://doi.org/10.1016/j.theriogenology.2010.01.003). Disagreement limit was considered to be values over 5%. As far as the authors are aware, there is no reported sensitivity and specificity of the method described.

L376: Please explain why the xylazine doses in four horses had a shorter biological half-life.

A comment was added in lines 393-394.

L492: I suggest that the authors revise the conclusions to align with the aims described in L72-78.

Conclusions were adapted accordingly.

L498-504: The points related to these aspects are missing.

This was now added.

Round 2

Reviewer 1 Report

Comments and Suggestions for Authors

Line 90: Replace “since” with “due to”

Line 100-101: How was PCV and blood L-lactate measured?

Line 101: Should read as “done prior to sedation”.

Line 146-150: How long after the instillation of local anesthetic did the scan begin? Was this the same time for lidocaine and mepivacaine? Was the time the same for every horse? This is important, as it would allow an equine vet to replicate your technique.

Line 225: Please remove “Error! Reference source not found” as rest of text appears correct.

Line 392-396: Do the authors feel that instillation of local anesthetic is essential for this procedure? If so, it should be stated in this paragraph.

Author Response

Line 90: Replace “since” with “due to”

This has been corrected.

Line 100-101: How was PCV and blood L-lactate measured?

PCV by centrifugation or by automated hematology analyser (Sysmex XN-100)

Blood L-lactate: was checked with StatStrip Xpress Lactate System with StatStrip Lactate Test Strips (Nova Biomedical, Waltham, MA, USA).

Information accordingly was added to the manuscript line 102-105.

Line 101: Should read as “done prior to sedation”.

This has been corrected.

Line 146-150: How long after the instillation of local anesthetic did the scan begin? Was this the same time for lidocaine and mepivacaine? Was the time the same for every horse? This is important, as it would allow an equine vet to replicate your technique.

This was specified further in the manuscript.

Line 225: Please remove “Error! Reference source not found” as rest of text appears correct.

This has been corrected.

Line 392-396: Do the authors feel that instillation of local anesthetic is essential for this procedure? If so, it should be stated in this paragraph.

A sentence was added line 397-400.

Reviewer 3 Report

Comments and Suggestions for Authors

Dear authors, thank you for taking the time to improve your paper! It is an important addition to the literature and I enjoyed reviewing it. 

Author Response

Dear Reviewer, Thank you very much for your kind words and thoughtful feedback. We sincerely appreciate the time and effort you dedicated to reviewing our work. We're glad to hear that you found the paper to be a valuable contribution to the literature.